# Short-Wavelength Infrared Imaging of Infected and Affected Dentin

**DOI:** 10.3390/diagnostics14070744

**Published:** 2024-03-30

**Authors:** Morgan Ng, Yi-Ching Ho, Spencer Wycoff, Yihua Zhu, Daniel Fried

**Affiliations:** 1Department of Preventive and Restorative Dental Sciences, University of California, San Francisco, 707 Parnassus Ave, San Francisco, CA 94143, USAspencer.wycoff@ucsf.edu (S.W.); yihua.zhu@ucsf.edu (Y.Z.); 2Department of Stomatology, Taipei Veterans General Hospital, Taipei 11217, Taiwan; ycho5@vghtpe.gov.tw

**Keywords:** SWIR imaging, caries detection, affected dentin, infected dentin

## Abstract

Stains produced by bacteria or those found in blood and food byproducts accumulate in highly porous caries lesions. They can interfere with accurate diagnosis and the selective removal of carious tissue during cavity preparations. Short-wavelength infrared (SWIR) imaging studies have shown that stain molecules do not absorb light beyond 1200 nm. The objective of this study was to image affected and infected dentin atSWIR wavelengths. Sections of 3 mm thickness were cut from the extracted teeth with deep dentinal lesions. The sound (normal), affected (stained), and infected (demineralized) dentin on each section were examined with reflected light at wavelengths from 400 to 1700 nm, red and green fluorescence, and with optical coherence tomography (OCT). Microcomputed tomography (microCT) was used to measure the mineral density at each location investigated. Significant (*p* < 0.05) differences were observed in the reflected light intensity at 400–850 nm and for fluorescence between the sound, affected, and infected dentin. SWIR imaging did not show significant reductions in reflectivity for the affected and infected dentin. SWIR images may be valuable for monitoring the lateral spread of dentinal lesions on the occlusal surfaces of teeth.

## 1. Introduction

Stains can profoundly influence the visual diagnosis of caries lesions, particularly occlusal lesions, and potentially lead to overtreatment [1,2,3]. They mask demineralization in the pits and fissures of the occlusal surfaces of teeth [4]. Stain molecules accumulate in dental plaque and cannot be completely removed from the pits and fissures by cleaning [5]. They are responsible for pigmentation in the visible range and do not absorb light beyond 1200 nm [6,7,8]. Subsurface staining in the underlying dentin can generate dark shadows that may suggest that the lesion has spread throughout the underlying dentin [9,10,11]. In the International Caries Detection and Assessment System (ICDAS) such an underlying dark shadow would yield an ICDAS score of four [12]. A recent study has shown that occlusal lesions with underlying dark shadows on the permanent teeth had a slow rate of progression [13]. In a previous clinical study involving short-wavelength infrared (SWIR) imaging, an occlusal caries lesion that had been scheduled for restoration showed a distinct underlying dark shadow from 400 to 700 nm (Figure 3 in ref. [14]). That dark shadow was not visible in the SWIR and OCT images, indicating that demineralization was confined to the fissure and had not spread throughout the underlying dentin, as suggested by the presence of the dark shadow [14]. Many dark shadows are likely due to the underlying affected or stained dentin rather than the underlying infected or demineralized dentin. Since stains do not absorb light within the SWIR range beyond 1200 nm, we hypothesize that SWIR imaging can be used to help differentiate between the affected and infected dentin. The areas of the occlusal surface that show dark shadows at visible wavelengths and do not show increased reflectivity within the SWIR range are likely areas with underlying affected dentin. In addition, we hypothesize that SWIR images better show the true spread of demineralization below the surface to aid in the more selective removal of infected dentin.

Affected dentin is defined as stained, but hard and intact, dentin, while infected dentin is soft and demineralized [15]. Many dentists still subscribe to the concept that all the infected dentin needs to be removed during a cavity preparation [15]. Some clinicians advocate for only partial or stepwise removal [16,17]. Methods have been developed to help differentiate between affected and infected dentin during caries excavation, including using caries indicator dyes [18] and soft burrs [19]. However, none of these methods have found widespread acceptance. SWIR imaging may also be valuable during caries excavation to help discriminate between the areas of affected and infected dentin. 

Several studies have demonstrated that demineralization can be imaged with higher contrast within the SWIR range than at visible and near-infrared (NIR) wavelengths [20]. The scattering coefficient of sound enamel is 20 to 30 times higher in the visible range versus the SWIR range at 1300 nm, providing high contrast of demineralization within the SWIR range [20]. Water absorption also influences lesion contrast, particularly for dentin, and there is a large increase in the contrast of demineralization on root surfaces at wavelengths beyond 1400 nm [21]. Hyperspectral images of Zakian et al. [22] showed that teeth appeared darker with increasing wavelengths due to increased water absorption. Novel imaging configurations, such as occlusal transillumination and cross-polarization reflectance imaging, can be used to image lesions within the SWIR range, and clinical studies have shown a much higher sensitivity than that of radiographs on both occlusal and proximal surfaces [23]. Hyperspectral SWIR imaging studies have also been carried out to identify caries lesions [22,24,25]. Multiple commercial NIR clinical imaging devices have been introduced, including CariVu (Dexis, Hatfield, PA, USA), which uses NIR occlusal transillumination with 780 nm light [26,27], and the Vistacam IX (PROXI) from Durr Dental (Bietigheim-Bissingen, Germany), which uses NIR reflectance at 850 nm [28]. Imaging at shorter NIR wavelengths near 830 nm was first investigated 20 years ago [23]. The 830 nm system was capable of a better performance than the visible systems, but the contrast was significantly lower than that attainable at 1300 nm, and the simulated lesions could not be imaged through the full enamel thickness due to greater light scattering [23], and the stains still absorbed at these wavelengths [4]. 

Two fluorescence (FL) imaging approaches have been introduced to aid in imaging dental caries that use “green” and “red” fluorescence [29,30,31,32,33,34,35,36,37,38,39,40,41]. In the green FL approach usually referred to as quantitative light fluorescence (QLF), blue-green light is used to excite collagen in the dentin, and the green FL is imaged. This method is really an FL loss method since demineralization attenuates the underlying green FL from the collagen in dentin, and the demineralized areas appear darker than the sound areas [42]. In the second method or the red FL method, blue-green or red light is used to excite the FL from porphyrin molecules that accumulates in the lesion areas [34]. Increased levels of porphyrins accumulate in dentinal caries lesions due to the high porosity. Red FL has been employed for caries detection, and the Diagnodent commercial device uses such fluorescence [36,37,38,40]. However, red FL is not specifically associated with cariogenic bacteria since they do not contain porphyrins, nor is it correlated with the degree of demineralization. In the case of green FL, stains can absorb the light and interfere, while for red FL, it is the stain molecules (porphyrins) that emit the FL.

The purpose of this study was to investigate the influence of subsurface staining in lesion areas within the SWIR range. The effect of stains on the contrast of occlusal lesions has been investigated previously, but this earlier study focused on extrinsic staining in the pits and fissures of the occlusal surfaces of whole teeth [4]. In this study, 3 mm thick sections were cut from lesions on coronal and root surfaces so that the influence of stain molecules penetrating deep into dentin could be investigated. We hypothesize that differences in the optical appearance of such stained lesion areas between the visible and SWIR range can be exploited to aid in the diagnosis of caries lesions and the selective removal of carious tissue during cavity preparations.

## 2. Materials and Methods

### 2.1. Tooth Sections

Teeth without identifiers were collected from dental clinics in the San Francisco Bay area. Such collection is exempt from IRB approval, Exemption 45 CFR 46.104 Category 4 from the U.S. Department of Health and Human Services (HHS) and is not considered Human Subjects research. The teeth were sterilized using gamma radiation and stored in 0.1% thymol solution to maintain tissue hydration and prevent bacterial growth. Teeth with suspected dentinal caries (*n* = 28) were sectioned into 3 mm thick slices, and the lesion was bisected using a linear precision saw, Isomet 2000 from Buehler (Lake Buff, IL, USA). Only those sections that had dentinal lesions that penetrated beyond the dentinal–enamel junction (DEJ) were selected, and only one section from each tooth was included in the study. Lesions were visually identified by the either the presence of a stain or whiter demineralization below the DEJ. All optical measurements were carried out with samples internally moist without excess water on the sample surfaces. Specific locations on each tooth section were chosen based on the mineral density measured with microCT and the presence of stains in the 400–700 nm color images. Locations representing sound enamel (SE), demineralized enamel (DE), sound dentin (SD), demineralized dentin (DD), and affected dentin (AD) were chosen. Sound areas with no mineral loss and visible staining were chosen, while demineralized areas with measurable mineral loss were chosen. Stained areas of demineralized enamel were intentionally avoided since only a few areas were discolored. Affected dentin locations that showed stains in the 400–700 nm images and had mineral densities higher than 1.8 as indicated by microCT were chosen. All demineralized areas in dentin showed stain accumulation in the 400–700 nm images; therefore, all chosen DD locations were stained as well as demineralized.

### 2.2. Microcomputed Tomography (microCT)

Sections were imaged using microCT with a 10 µm resolution. A Scanco MicroCT 50 from Scanco USA (Wayne, PA, USA) was used to acquire images of each section. Acquisition parameters used for the microCT images were 90 kVP, 200 mA, 18 W, 20 FOV, 10 µm voxel size, 500 ms integration time, and an aluminum 0.5 mm filter. Lesion mineral density measurements were carried out using Dragonfly from ORS (Montreal, QC, Canada). Imaging was carried out with the samples immersed in water to prevent shrinkage and protect the samples from thermal damage during imaging. The mean density in g/cm^3^ was calculated for each chosen location that consisted of a rectangular prism of dimensions 400 × 400 × 500 μm.

### 2.3. Optical Coherence Tomography (OCT)

An IVS-2000-HR OCT system from Santec (Komaki, Aichi, Japan) was used for this study. This system utilizes a swept laser source and a handpiece with a microelectromechanical (MEMS) scanning mirror and imaging optics. It can acquire complete tomographic images of a volume of 5 × 5 × 5 mm in approximately 3 s. The body of the handpiece is 7 × 18 cm with an imaging tip that is 4 cm long and 1.5 cm across. This system operates at a wavelength of 1312 nm, with a bandwidth of 173 nm, with a measured depth resolution in air of 8.8 µm (3 dB). The lateral resolution is 30 µm (1/e^2^) with a measured imaging depth of 5 mm in air. The integrated reflectivity (ΔR) at each location was calculated by taking the sum of the mean reflectivity of each 400-by-400 μm slice over a depth of 500 μm.

### 2.4. Reflectance Measurements from 400 to 1750 nm

Visible color images of the samples were acquired using a USB microscope, Model AM7915MZT from BigC, with an extended depth of field and cross polarization. The digital microscope captured 5 mega-pixel (2952 × 1944) color images. The RGB 24-bit color images were converted to 8-bit grayscale images using the formula 0.2989 × (R) + 0.5870 × (G) + 0.1140 × (B) in MATLAB from Mathworks (Natick, MA, USA) for intensity calculations. 

A DMK-3002–IR near-IR sensitive CCD camera (Imaging Source, Charlotte, NC, USA) equipped with an Infinity (Centennial, CO, USA) Infinimite lens was used to acquire 850 nm images. A bandpass filter centered at 850 nm with a bandwidth of 90 nm was used. 

A high-sensitivity InGaAs focal plane array camera, Model GA1280J from Sensors Unlimited (Princeton, NK, USA), with 1280 × 1024 pixels and a 15 μm pixel pitch equipped with an InfiniMini lens from Infinity (Centennial, CO, USA) and an additional achromatic lens, f = 60-mm, was used to acquire images from 1300 to 1750 nm. A stabilized Tungsten IR light source, Model SLS202, from Thorlabs (Newton, NJ, USA), with a peak output at 1500 nm and collimating optics, was the light source. Three bandpass filters, wavelengths (bandwidths) centered at 1300 (90), 1460 (85), and 1675 (90) nm, were used. The mean reflectivity was measured for each location that consisted of a 400 μm square area matched to the microCT and OCT positions.

For 400–700 and 850 nm reflectance measurements, the intensity range or bit depth of the sensor was 256 (8-bit), while for SWIR imaging, it was 4096 (12-bit). Therefore, for all the reflectance measurements, the background intensity was subtracted, and the intensity was divided by the maximum intensity or bit depth of each sensor to normalize the intensity among the three sensors. 

### 2.5. Fluorescence (FL) Measurements

Green light FL images were acquired using a USB microscope with 5 blue (480 nm) light emitting diodes (LEDs) and a 510 nm longpass filter, Model AM4115TW-GFBW from BigC (Torrance, CA, USA). Red FL images were acquired using a second USB microscope with 5 red (620 nm) excitation LEDs and a 650 nm longpass filter, Model AM4115T-DFRW from BigC, designed to image porphyrin fluorescence. For intensity calculations, for green FL, only an 8-bit green image was extracted from the RGB 24-bit color images. For red FL, only an 8-bit red image was extracted from the 24-bit color images. The mean FL intensity was calculated for each location that consisted of a 400 μm square area matched to the microCT and OCT positions. The background intensity was subtracted from each measurement, and the intensity was divided by 256. 

### 2.6. Statistics

Most groups (75%) were normally distributed; therefore, non-parametric statistics were used. Repeated measurements of one-way analysis of variance (ANOVA) followed by the Tukey–Kramer post hoc multiple comparison test were used to compare the intensities of sound, affected, and demineralized dentin. The enamel intensities were compared using a paired *t*-test (two sided). Statistical calculations were performed with Prism 6 from Graphpad Software (Boston, MA, USA).

## 3. Results

Images from three of the twenty-eight sections are shown in Figure 1, Figure 2 and Figure 3. Figure 1 shows an occlusal lesion with heavy staining that appears to penetrate halfway through the dentin towards the pulp chamber based on the color image (400–700 nm). However, microCT shows that demineralization is limited to just below the dentinal–enamel junction (DEJ). In the color image, Figure 1A, there is a highly stained, darker region along with a more lightly stained region deeper into the dentin. MicroCT, Figure 1G, shows that demineralization is only present in the very dark stained area of the dentin. There are six colored markers added to Figure 1A, showing the positions where the measurements were taken for this section; white corresponds to the background, black to the sound enamel and dentin, red to the demineralized enamel, yellow to the demineralized dentin, and blue to the affected dentin. Matching markers are shown in the microCT image, Figure 1G, except for the background since there was no background subtraction for microCT. In the green FL image, Figure 1B, the darker areas show higher levels of demineralization. However, the stained areas of dentin that are not demineralized also appear darker, including the area that corresponds to the position of the blue marker in Figure 1A,G which correspond to the affected dentin that is not demineralized. Note that the lesion contrast in the green FL image is inverted since it represents FL loss, i.e., demineralized areas should appear darker instead of lighter in green FL images. The red FL image shown in Figure 1C shows the accumulation of porphyrins and indicates the full extent of the penetration of the stain throughout the dentin. There is increased red FL all the way to the pulp chamber, and the area of increased red FL encompasses a much larger area than the areas in which demineralization is indicated with microCT. Figure 1D,E, and F shows reflected light images at 850, 1300, and 1460 nm. Most of the 1675 nm images appear very similar to the 1460 nm images and are not shown in the figures. In the 850 nm image, the demineralized enamel has a higher reflected light intensity (whiter) than the sound enamel. In the dentin, the area that is highly stained and very dark in the color image also appears very dark. However, the affected/demineralized dentin area to the left of the large crack in the dentin only appears slightly darker than the sound dentin to the right of the crack. At 1300 nm, where none of the stain absorbs light, the area that appears very dark at 850 nm appears much smaller, and the difference in the intensity between the affected/demineralized dentin to the left of the crack and the sound dentin to the right of the crack appears larger. In addition, differences in intensity appear between two areas of the affected/demineralized dentin to the left of the crack. An upper lighter area is indicated by the orange marker in Figure 1F, and a less dark area is indicated by the purple marker. The density of the orange marker is 1.8, while that of the lower marker is 2.1. The density in the sound dentin to the right of the crack at the position of black cross is also 2.1. This higher mineral density can explain why the lower area appears darker or more transparent within the SWIR range. At 1460 nm, the difference in intensity between the affected/demineralized dentin area to the left of the large crack and the normal dentin to the right of the crack appears greater than at 850 and 1300 nm wavelengths. The intensity differences between the areas of dentin indicated by the orange and purple markers in Figure 1F are also more pronounced than those in the 850 and 1300 nm images. The OCT image of integrated reflectivity at 1300 nm shown in Figure 1H also shows high contrast between the two areas of dentin indicated by the orange and purple markers on the left side of the crack. 

The occlusal lesion shown in Figure 2 is more severe, and demineralization penetrates more than halfway through the dentin between the DEJ and the pulp chamber. The color image, Figure 2A, shows three distinct areas of dentin. There is an inner stained region that seems to match the area of demineralization shown in the microCT image, Figure 2G, except for the area to the right of the fissure just below the DEJ and the area where the blue marker is located. There is a second whiter area surrounding the stained area, followed by a slightly darker translucent area below that. The green FL image in Figure 2B shows a larger area of FL loss than the area of demineralization indicated in the microCT image and does not show any FL loss to the right of the central fissure. The red FL image, Figure 2C, shows a larger area of increased FL that extends deeper than the area of demineralization indicated in microCT, and that area of increased FL also extends to the right of the central fissure in a similar fashion to the color image, Figure 2C. The 850, 1300, and 1460 nm images all show increased reflectivity in the dentin that closely matches the demineralization shown in the microCT image. The second band of dentin in the color image that appears much whiter than the inner band where demineralization is present appears darker than the inner band at the 850–1460 nm wavelengths. The contrast between the sound/translucent dentin and the demineralized dentin appears to be the highest at 1460 nm. The OCT image shown in Figure 2H does not seem to show as much contrast between the demineralized and sound/translucent dentin.

The third example shown in Figure 3 shows two occlusal lesions with unusual structures in both the enamel and dentin lesion areas. The lesion in the central fissure shows a U-shaped dark (stained) region in the enamel in the color image, Figure 3A, that is also visible in the microCT image, Figure 3G, as a highly mineralized zone. The color image shows a band of highly stained dentin along the DEJ below both the fissures and a large lightly stained area below that, which penetrates halfway to the pulp chamber. Where the stain terminates, there is a thin band of what appears to be translucent dentin. The green FL image in Figure 3B shows FL loss that mirrors the staining in the color image. The red FL shown in Figure 3C is very strong and encompasses a large area, penetrating all the way to the pulp chamber. The 850, 1300, and 1460 nm images also show the dark band in the enamel in the lesion of the central fissure. The demineralized dentin near the base of the right fissure has a higher reflectivity in all three images. The highly stained band near the DEJ in the color image also appears dark in the three NIR/SWIR images and appears very dark in the 1460 nm images and connects with the band of dark translucent dentin located below the stained dentin. The OCT image shown in Figure 3H also shows areas of translucent dentin.

The mean (s.d.) values of the reflected light intensities for 400–1700 nm and fluorescence, the integrated reflectivity from OCT, and the mineral densities from microCT for *n* = 17 enamel and *n* = 28 dentin sites on the 28 tooth sections are tabulated in Table 1. For enamel, the mean reflectivity was higher for the demineralized enamel than for the sound enamel, and all the measurements showed significant differences between the selected sound and demineralized areas, except for reflectance from 400 to 700 nm. It is important to note that all the enamel demineralized areas that were chosen were not stained. Even though enamel demineralized areas were chosen that did not show discoloration in the reflected light color images, there was still a significant increase in red FL from those areas.

The sound dentin yielded the highest reflectance at 400–700 nm, followed by the affected and demineralized dentin with significant differences between all three groups. Since these are measurements of reflectance, the reflectivity of demineralized dentin was expected to be higher than that for the sound dentin, assuming there was no absorption by the stains. At 850 nm, the same progression of intensities was observed. However, the values for the sound and affected dentin were statistically similar, and only the intensity for the demineralized dentin was significantly lower. For the SWIR measurements at 1300, 1460, and 1675 nm and the integrated reflectivity from OCT, all the mean intensities were statistically similar for the three dentin locations.

For the intensity of red FL, the affected dentin had the highest intensity, followed by the demineralized and sound dentin. All the groups were significantly different. Higher levels of red FL were anticipated from the affected and demineralized dentin compared to those from the sound dentin. For the intensity of green FL, the sound dentin had the highest intensity, followed by the affected and demineralized dentin. All the groups were significantly different. Less green FL was anticipated from the demineralized dentin compared to that from the sound dentin, assuming there was no interference from the stains.

The mean mineral densities measured using microCT were significantly different between the sound and demineralized enamel and between the sound and demineralized dentin. The mineral densities between the sound and affected dentin were statistically similar.

## 4. Discussion

In demineralized dental hard tissues, pores are created from mineral loss that increases the amount of light scattering. Therefore, in the absence of any light absorption by stains, there should be a large increase in reflectivity with demineralization, and this has been observed in simulated lesions produced on enamel and dentin surfaces [43,44]. In this study, there was a large difference in the mean mineral densities between the sound and demineralized enamel locations, 3.0 vs. 2.0 g/cm^3^, as indicated by microCT. However, there was no significant difference in the reflected light intensity from enamel at 400–700 nm, even after intentionally avoiding the discolored locations. This suggests that even though the areas in enamel chosen for measurement were not discolored, the stain molecules in the lesion still absorbed enough visible light to greatly reduce the reflectivity from the enamel lesion areas. There was substantial red FL from the enamel areas that appeared sound, which confirms that the stain molecules had accumulated in those areas.

The influence of stains on the reflectivity of the more porous dentin was even greater than that for the enamel from 400 to 700 nm. The mean reflectivity of the sound dentin was three times higher than the reflectivity of the demineralized dentin. Reduced reflectivity is expected due to light absorption by the stains at wavelengths less than 1200 nm, and all the demineralized dentin locations showed discoloration. In addition, the reflectivity is also expected to decrease in areas of translucent, sclerotic, or reparative dentin. In these areas, minerals fill the dentinal tubules, which decrease the amount of light scattering, and the dentin becomes more transparent. Reparative dentin formation occurs as a response to caries, and it also occurs with increasing age [15]. At 850 nm, the sound and affected dentin had a similar reflectivity that was slightly higher than that for the demineralized dentin. For the longer 1300, 1460 and 1675 nm wavelengths, the reflected light intensities were similar. It was anticipated, based on prior studies, that within the SWIR range, the sound and affected dentin would have similar intensities and that the demineralized dentin would have a much higher reflectivity than the sound dentin. The lightly stained tooth section shown in Figure 2 with the large dentinal lesion does indeed conform to this expected trend and shows increased reflectance of the demineralized dentin in the SWIR images. However, the areas of dentin demineralization appear darker for the other sections shown in Figure 1 and Figure 3. The selected area of demineralization in Figure 2 shows more demineralization, with a density of 1.2, while the other two areas show densities of 1.3 and 1.4 for Figure 1 and Figure 3, respectively. In an imaging study of the reflectivity of root surfaces at the 850, 1300, 1460, and 1675 nm wavelengths on twenty intact extracted teeth, the contrast of the root caries lesions and dental calculus was markedly higher at wavelengths of 1460 and 1675 nm compared to 1300 nm, while the lesion contrast at 400–850 nm was negative due to heavy stain accumulation [21]. In addition, in several studies, including clinical studies employing CP-OCT and OCT at 1300 nm, demineralization on the root surfaces is clearly visible with increased reflectivity [21,44,45]. Moreover, in the OCT images acquired from occlusal surfaces, those occlusal lesions that have spread laterally below the DEJ and for the proximal lesions that have reached the dentin, there is increased reflectivity at the DEJ [14,46,47]. It is not clear why some of the demineralized areas of dentin appear dark within the SWIR range. The areas of demineralization in Figure 1 and Figure 3 also appear dark in the red FL images. It may be possible that there is some byproduct that is produced via the breakdown of the organic matrix of dentin, such as carbon that absorbs light. We know that as dentin chars during Nd:YAG laser irradiation at 1064 nm, there is a large rise in light absorption as the collagen is carbonized [48]. Such an absorber could absorb both the red FL and light within the SWIR range, causing these areas to appear darker.

We anticipated that the differences in mineral content between the sound dentin and those of the translucent, sclerotic, or reparative dentin would be large enough to easily monitor with microCT because the dentinal tubules fill with minerals. However, the differences appear to be slight, and microCT was most useful for showing areas of demineralization in the dentin. There appears to be a very large variation in the transparency of the “sound” dentin, particularly at longer wavelengths. The standard deviations for the reflectivity of sound enamel are much larger at 850–1675 nm than they are for 400–700 nm. Even though translucent dentin is visible from 400 to 700 nm, at longer wavelengths, the translucent dentin can appear completely transparent and appears to have a similar reflectivity to that of the sound enamel. Light scattering in sound enamel decreases markedly with increasing wavelengths because the light scattering centers in the small enamel crystals behave as Rayleigh scatterers that are much smaller than the wavelength of light [49]. In contrast, light scattering in sound dentin is dominated by the dentinal tubules that are similar in size to the wavelength of light [49,50,51]. Therefore, light scattering in dentin does not decrease markedly with increasing wavelength in a similar fashion to that in enamel [49]. However, as minerals fill the dentinal tubules, we expect the light scattering in dentin to behave in a more similar fashion to enamel since the dentinal tubules are no longer the principal scatterers. It appears that there are large differences in the light scattering of the translucent, sclerotic, and reparative dentin that can be resolved within the SWIR range that are not as apparent at shorter wavelengths, for example, the three areas of dentin at the positions of the two triangles and the black cross in Figure 1F. Even though microCT shows little difference in mineral density between the three positions, there are large differences in reflectivity within the SWIR range.

At wavelengths beyond 1400 nm, water absorption increases and plays a role in the contrast of demineralized dentin [21]. The contrast of root caries lesions increased only up to 1400 nm and did not increase significantly with increasing wavelength, as opposed to lesions on enamel surfaces, where the contrast continues to increase with increasing wavelength [21]. This suggests that water absorption has a greater significance for the reflectivity of dentin, and the water content of sound dentin is much higher than that for the sound enamel.

The green and red FL images are both valuable for viewing the differences between the sound, demineralized, and affected dentin. Green FL was highest for the sound dentin, which has the most intact collagen. However, the interpretation of green FL loss is problematic in that it is not clear that FL is reduced by increased light scattering due to demineralization or absorption by the stain molecules. The affected dentin showed a large and significant reduction in green FL due to absorption by the stains. Red FL due to porphyrins was very effective in showing the penetration of stains throughout the dentin. The affected dentin exhibited greater FL than the demineralized or infected dentin, showing that large amounts of stain molecules can become trapped as new mineral fills the dentinal tubules in those areas. Intense red FL emissions appear in the areas of dentin that appear almost completely transparent in the SWIR images. Many areas of increased red FL are far larger than the areas of demineralized dentin, and red FL appears in many areas that appear sound and are not discolored. The areas of red FL in Figure 1 and Figure 3 show increased red FL all the way to the pulp chamber. However, microCT shows that demineralization is limited to near the DEJ. Red FL has been proposed as a means of guiding caries removal using both laser and mechanical means [43,52,53,54]. This study shows that the poor localization of red FL indicates that it is not well suited for the guided removal of caries lesions and shows how it may lead to false positive readings.

Many of the lesions in both the enamel and dentin had complex lesion structures consisting of highly mineralized translucent areas within the demineralized areas, such as the enamel lesion in the central fissure or the dark bands in the dentin in Figure 3. The complexity of such structures can be easily used to explain the challenge of using soft dental burrs and the mixed penetration of caries indicator dyes. SWIR and OCT images may be useful to help guide caries removal by more accurately showing the lateral spread of caries lesions below the enamel. Other studies have shown that SWIR and OCT images can be used to help guide the selective removal of calculus from root surfaces [55]. Serial SWIR images have been used to facilitate the selective removal of small occlusal lesions confined to the enamel [56]. We had hoped to observe a large increase in the mean reflectivity of infected dentin compared to that of sound dentin so that it would be straightforward to use the same approach to remove dentinal lesions. It appears that this is more complex, and additional studies are needed to determine if serial SWIR or OCT images taken in real time during cavity preparations can be used for the selective removal of the deep dentinal lesions investigated in this study.

It is important to explain the clinical significance of the mean reflectivity of the affected and sound dentin being similar within the SWIR range, while the reflectivity of the affected dentin is significantly lower than that of the sound dentin at visible wavelengths. Visible wavelengths cannot be used to determine whether shadows are caused by affected or infected dentin. However, SWIR imaging and OCT can be potentially used to eliminate those shadows caused solely by affected dentin or only by stains. Large areas of affected dentin are unlikely to create subsurface shadows within the SWIR range similar to what is observed at visible wavelengths. There are profound differences in the way reduced light scattering due to the formation of translucent dentin and reduced reflectivity due to absorption by stains translate to the visibility of lesion shadows viewed from the occlusal surface. Much of the light that is reflected from the tooth surfaces arises from the underlying sound dentin that has a higher reflectivity than the sound translucent enamel. This study shows that the affected and demineralized dentin had markedly lower reflectivity than the sound dentin at 400–700 nm. Light scattering from any sound dentin underlying the affected or demineralized dentin areas would also be expected to be absorbed by that stain before exiting the occlusal surface. Therefore, the large decrease in reflectivity caused by the demineralized and affected dentin would likely cause areas of demineralized and affected dentin underneath sound enamel peripheral to occlusal pits and fissures to appear as dark shadows at 400–700 nm. Based on the significantly lower reflectivity of the affected dentin compared to that of the sound dentin, it is not feasible to determine if that shadow is caused by the demineralized dentin or by only the affected dentin. Within the SWIR range, there was no significant reduction in the reflectivity of the demineralized and affected dentin compared to that of the sound dentin. Moreover, if the affected dentin in those areas appeared darker than the surrounding sound enamel, it was due to increased transparency rather than by absorption by the stain. Light scattering from any sound dentin underlying the affected transparent dentin would still pass through the more-transparent affected dentin unattenuated and such affected areas would not appear darker from the occlusal surface. This explains why the large shadow of the occlusal caries lesion that was visible from 400 to 700 nm (Figure 3 in a previous study [14]) was not visible in the SWIR and OCT images. In addition, dentinal lesions spread laterally along the DEJ and both the enamel and dentin near the DEJ typically undergo demineralization. Therefore, it is likely that such demineralized areas would still be visible with increased reflectivity from the occlusal surface in the 1300 nm OCT and SWIR images [14,46,47], even if the demineralized dentin below the DEJ has lower reflectivity than the surrounding sound dentin and would not appear darker. One limitation of this study is that extracted teeth with visible shadows were not used. This was due to the challenge in collecting a sufficient number of samples. We hope to be able to collect enough extracted teeth with visible shadows for use in a future SWIR imaging study. It would also be useful to examine the areas of infected dentin that had reduced reflectivity within the SWIR range with infrared spectroscopy to determine if there is increased absorption and identify what chemical species are responsible.

## 5. Conclusions

Multispectral reflected light and fluorescence images are valuable for showing the differences between sound, demineralized, and affected dentin. There is a large and significant decrease in the reflectivity of demineralized and affected dentin compared to that of sound dentin at 400–700 nm. Clinicians should express caution upon treating occlusal lesions with shadows, since shadows may not be a reliable indicator of severe dentinal lesions, since the shadow may be caused solely by subsurface stains that encompass a much larger area than the actual lesion. SWIR imaging may be more reliable for monitoring the lateral spread of lesions peripheral to the occlusal pits and fissures since subsurface stains do not absorb light as strongly as they do in the visible range.

## Figures and Tables

**Figure 1 diagnostics-14-00744-f001:**
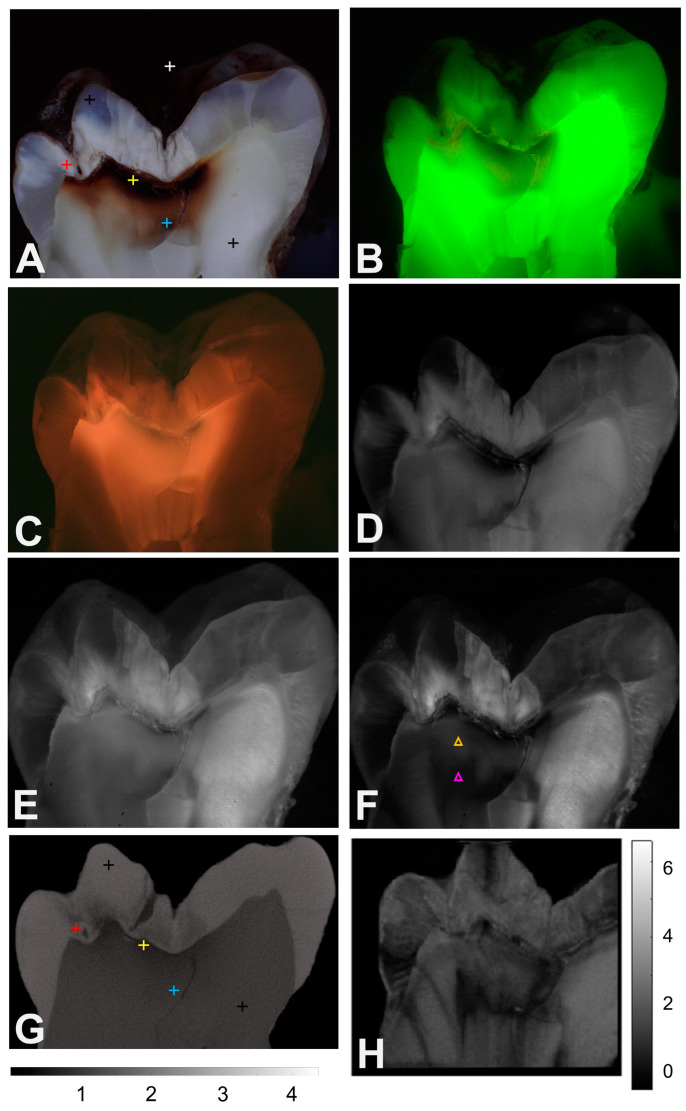
(**A**) Color (400–700 nm), (**B**) green FL, (**C**) red FL, (**D**) 850 nm, (**E**) 1300 nm, (**F**) 1460 nm, (**G**) microCT, and (**H**) OCT images of one of the tooth sections with an occlusal lesion, with very heavily stained areas in the dentin. The positions where measurements were taken are indicated by the colored markers in (**A**,**G**); white is the background, black is the sound enamel or dentin, red is the demineralized enamel, yellow is the demineralized dentin, and blue is the affected dentin. The intensity scale in (**G**) is the density, and the scale in (**H**) is the integrated reflectivity ×10^4^. The orange and purple markers in (**F**) represent two areas with densities of 1.8 and 2.1, respectively.

**Figure 2 diagnostics-14-00744-f002:**
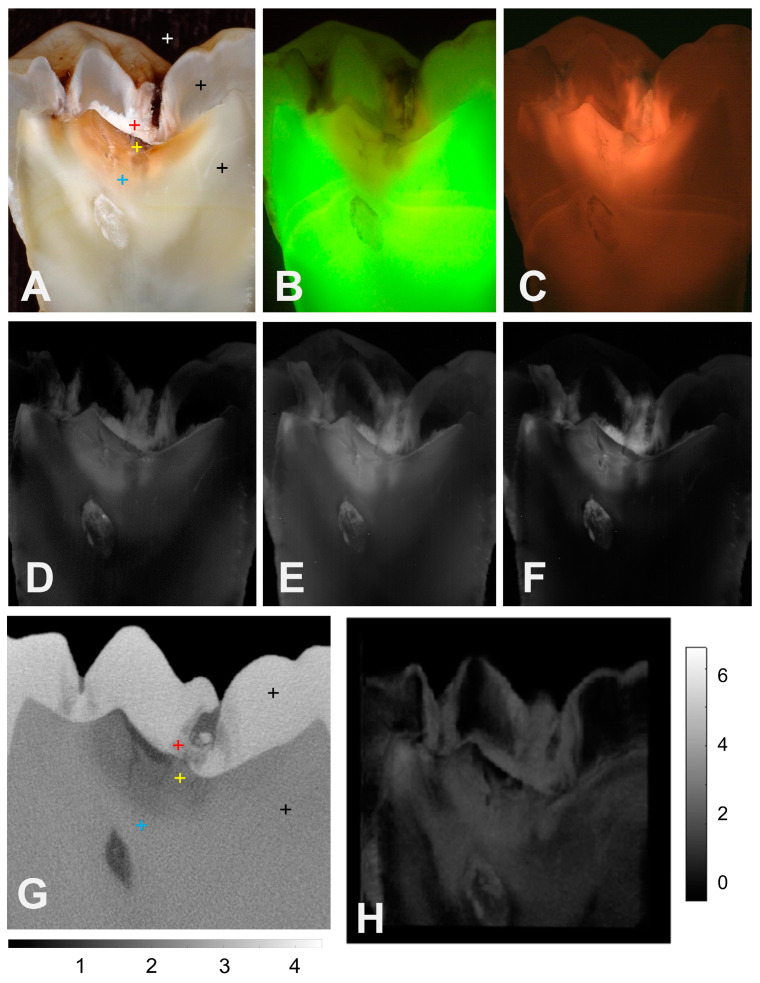
(**A**) Color (400–700 nm), (**B**) green FL, (**C**) red FL, (**D**) 850 nm, (**E**) 1300 nm, (**F**) 1460 nm, (**G**) microCT, and (**H**) OCT images of one of the tooth sections with an occlusal lesion that penetrates more than halfway to the pulp chamber that shows only minor stain accumulation. The positions where the measurements were taken are indicated by the colored markers in (**A**,**G**); white is the background, black is the sound enamel or dentin, red is the demineralized enamel, yellow is the demineralized dentin, and blue is the affected dentin. The intensity scale in (**G**) is the density, and the scale in (**H**) is the integrated reflectivity ×10^4^.

**Figure 3 diagnostics-14-00744-f003:**
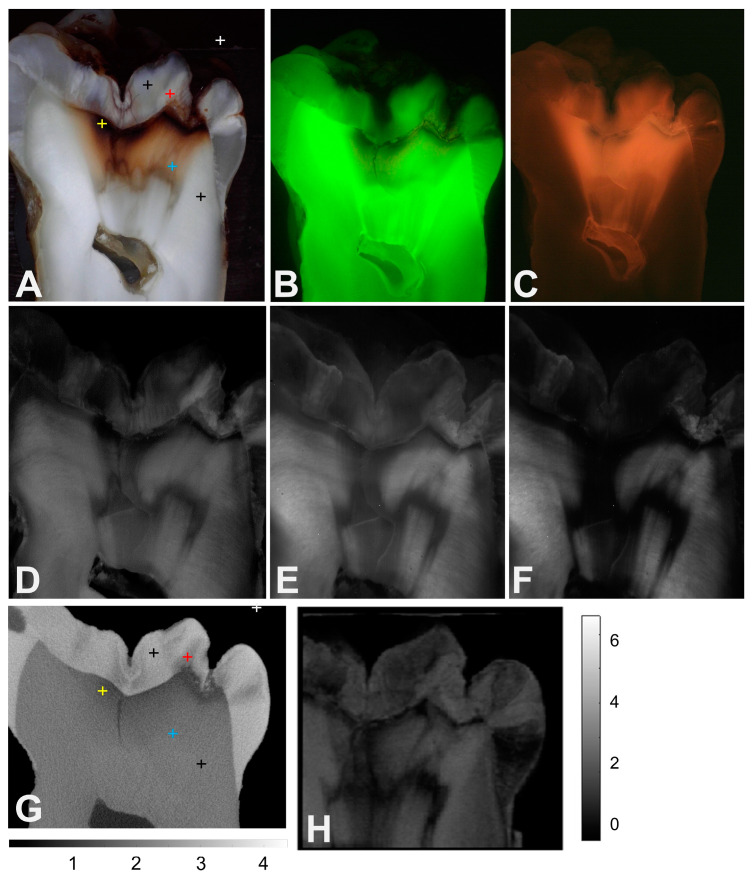
(**A**) Color (400–700 nm), (**B**) green FL, (**C**) red FL, (**D**) 850 nm, (**E**) 1300 nm, (**F**) 1460 nm, (**G**) microCT, and (**H**) OCT images of one of the tooth sections with an occlusal lesion that shows complex structures of translucent enamel and dentin in lesion areas. The positions where measurements were taken are indicated by the colored markers in (**A**,**G**); white is the background, black is the sound enamel or dentin, red is the demineralized enamel, yellow is the demineralized dentin, and blue is the affected dentin. The intensity scale in (**G**) is the density, and the scale in (**H**) is the integrated reflectivity ×10^4^.

**Table 1 diagnostics-14-00744-t001:** This table shows data for *n* = 17 enamel areas and *n* = 28 dentin areas. Sound enamel (SE), demineralized enamel (DE), sound dentin (SD), demineralized dentin (DD), and affected dentin (AD). An asterisk indicates a significant difference (*p* < 0.05) between SE and DE in each row. Columns with the same letter for SD, DD, and AD in each row are statistically similar (*p* > 0.05).

	SE	DE	SD	DD	AD
**400–700 nm**	0.52 (0.079)	0.58 (0.021)	0.75 (0.099) ^a^	0.27 (0.20) ^b^	0.47 (0.16) ^c^
**850 nm**	0.19 (0.12)	0.30 (0.17) *	0.32 (0.12) ^a^	0.25 (0.15) ^b^	0.30 (0.097) ^a^
**1300 nm**	0.15 (0.056)	0.33 (0.11) *	0.29 (0.12)	0.28 (0.084)	0.25 (0.097)
**1460 nm**	0.081 (0.047)	0.29 (0.17) *	0.20 (0.13)	0.17 (0.087)	0.14 (0.094)
**1675 nm**	0.063 (0.033)	0.25 (0.14) *	0.17 (0.11)	0.16 (0.072)	0.13 (0.073)
**Red FL**	0.076 (0.0015)	0.29 (0.16) *	0.21 (0.10) ^a^	0.42 (0.20) ^b^	0.55 (0.19) ^c^
**Green FL**	0.49 (0.13)	0.30 (0.15) *	0.81 (0.15) ^a^	0.32 (0.22) ^b^	0.65 (0.19) ^c^
**OCT (** **ΔR) × 10^3^**	8.6 (3.9)	17 (4.2) *	13 (4.7)	13 (4.4)	13 (4.9)
**MicroCT**	3.0 (0.039)	2.0 (0.45) *	2.1 (0.080) ^a^	1.4 (0.24) ^b^	2.0 (0.099) ^a^

## Data Availability

Data are available upon request.

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
