# Peer review of "Short-Wavelength Infrared Imaging of Infected and Affected Dentin"

_diagnostics, 2024, doi:10.3390/diagnostics14070744_

Round 1
Reviewer 1 Report
Comments and Suggestions for Authors
The article is very well written. The study presents valuable insights into using SWIR imaging for dental diagnostics with a well-structured methodology. I would recommend some points that could improve its impact and readability.
The introduction provides a comprehensive background. While it mentions the limitations of current diagnostic techniques, it could better emphasize how SWIR imaging specifically addresses these limitations.
In the Methodology section, please elaborate on the selection criteria for the teeth used to ensure the study's replicability.
The results are presented with corresponding figures and extensive descriptive text.
The discussion comprehensively addresses the study's findings; however, a detailed discussion of the implications of these findings for clinical practice would be beneficial.
Based on the study's results, the conclusions could be strengthened by offering more concrete suggestions for potential modifications in clinical practice.
Author Response
We have responded to the reviewer requests and comments to the best of our ability. Our detailed response is indicated below. We greatly appreciate the efforts of the reviewers to improve the manuscript and we hope we have sufficiently addressed their concerns. Our response is in italics and text added to the manuscript is highlighted in bold.
REVIEWER 1
The article is very well written. The study presents valuable insights into using SWIR imaging for dental diagnostics with a well-structured methodology. I would recommend some points that could improve its impact and readability.
The introduction provides a comprehensive background. While it mentions the limitations of current diagnostic techniques, it could better emphasize how SWIR imaging specifically addresses these limitations.
- Added new sentences at line 45 & 54 highlighting use of SWIR to aid in determining if the dark shadows visible on tooth surfaces at visible wavelengths is due to underlying affected (stained) or infected (demineralized dentin)
Since stains do not absorb light at SWIR wavelengths beyond 1200 nm, we hypothesize that SWIR imaging can be used to help differentiate between affected and infected dentin. Areas of the occlusal surface that show dark shadows at visible wavelengths and do not show increased reflectivity at SWIR wavelengths are likely areas with underlying affected dentin. In addition, we hypothesize that SWIR images better show the true spread of demineralization below the surface to aid in more selective removal of infected dentin.
SWIR imaging may also be valuable during caries excavation to help discriminate between areas of affected and infected dentin.
In the Methodology section, please elaborate on the selection criteria for the teeth used to ensure the study's replicability.
- Added line 94:
Only those sections that had dentinal lesions that penetrated beyond the dentinal enamel junction (DEJ) were selected and only one section from each tooth was included in the study. Lesions were visually identified by the either the presence of stain or whiter demineralization below the DEJ.
The results are presented with corresponding figures and extensive descriptive text.
The discussion comprehensively addresses the study's findings; however, a detailed discussion of the implications of these findings for clinical practice would be beneficial.
- Added line 376:
We had hoped to observe a large increase in the mean reflectivity of infected dentin compared to sound dentin so that it would be straightforward to use the same approach to remove dentinal lesions. It appears that this is more complex and additional studies are needed to determine if serial SWIR or OCT images taken in real-time during cavity preparations can be used for the selective removal of the deep dentinal lesions investigated in this study.
It is important to explain the clinical significance that the mean reflectivity of affected and sound dentin are similar at SWIR wavelengths while the reflectivity of affected dentin is significantly lower than sound dentin at visible wavelengths. Visible wavelengths cannot be used to determine whether the shadows are caused by affected and infected dentin, however SWIR imaging and OCT can be potentially used to eliminate those shadows caused solely by affected dentin or only by stains. Large areas of affected dentin are unlikely to create subsurface shadows at SWIR wavelengths similar to what is observed at visible wavelengths. There are profound differences in the way reduced light scattering due to the formation of translucent dentin and reduced reflectivity due to absorption by stains translate to the visibility of lesion shadows viewed from the occlusal surface.
Based on the study's results, the conclusions could be strengthened by offering more concrete suggestions for potential modifications in clinical practice.
- Added line 415:
Clinicians should express caution upon treating occlusal lesions with shadows, since shadows may not be reliable indicators of severe dentinal lesions, since the shadow may be caused solely by subsurface stains that encompass a much larger area than the actual lesion. SWIR imaging may be more reliable for monitoring the lateral spread of lesions peripheral to occlusal pits and fissures since subsurface stains do not absorb light as strongly as in the visible range.
Reviewer 2 Report
Comments and Suggestions for Authors
Please, revise lines 50-78 and 108-115, due to increased similarity.
Author Response
Please, revise lines 50-78 and 108-115, due to increased similarity.
The first section 50-78 is part of the introduction and the 2nd section describes the methodology for OCT. There is no overlap of the two sections. It is possible the line numbers viewed by reviewer 2 don’t match the original submission. We have reviewed the discussion and methodology sections and we do not see any obviously redundant sections.
Reviewer 3 Report
Comments and Suggestions for Authors
DEAR AUTHORS,
Thank you for the opportunity to review this interesting study.
In the era of minimally invasive dentistry, it is necessary to favor conservative approaches that allow the hard tissues of the tooth to be preserved. the development and clinical application of technological tools can and must make a difference in our daily clinical practice
YOU WILL FIND SOME OF MY COMMENTS BELOW:
line 34: surgical intervention... the term "surgical" is not appropriate
caries lesions with ICDAS scores > 4 do not necessarily require surgical treatment, but endodontic treatment
Line 38, 40, 54. There are acronyms for the first time in the text without the extended description.
line 101 please add the ethics committee approval number
Please analyze and better describe the limitations of the study
I found a large number of self-citations, are they all really necessary???? Please remove the superfluous self-citations or give me an explanation for the use of each of them
References 16, 19,28,40,45,52: please bold the year of publication
Authors name: please add a comma between Yi-Ching Ho and Spencer Wycoff
Author Response
Thank you for the opportunity to review this interesting study.
In the era of minimally invasive dentistry, it is necessary to favor conservative approaches that allow the hard tissues of the tooth to be preserved. the development and clinical application of technological tools can and must make a difference in our daily clinical practice
YOU WILL FIND SOME OF MY COMMENTS BELOW:
line 34: surgical intervention... the term "surgical" is not appropriate
caries lesions with ICDAS scores > 4 do not necessarily require surgical treatment, but endodontic treatment
- Removed “and that surgical intervention may be warranted”
Line 38, 40, 54. There are acronyms for the first time in the text without the extended description.
- Corrected
line 101 please add the ethics committee approval number
- No specific IRB is required. We are permitted by the UCSF Committee on Human Research to collect Human teeth without identifiers (NIH Exemption Category 4)
Please analyze and better describe the limitations of the study
- Added line 406:
One limitation of this study is that extracted teeth with visible shadows were not used. This was due to the challenge in collecting a sufficient number of samples. We hope to be able to collect enough extracted teeth with visible shadows for use in a future SWIR imaging study. It would also be useful to examine the areas of infected dentin that had reduced reflectivity at SWIR wavelengths with infrared spectroscopy to determine if there is increased absorption and identify what chemical species are responsible.
I found a large number of self-citations, are they all really necessary???? Please remove the superfluous self-citations or give me an explanation for the use of each of them
- We have replaced ten of our past citations from the introduction dealing with the introduction of SWIR reflectance and transillumination imaging with two review articles, refs. 20 & 23 reducing the number of self-citations by eight.
References 16, 19,28,40,45,52: please bold the year of publication
- Corrected
Authors name: please add a comma between Yi-Ching Ho and Spencer Wycoff
- Corrected
Reviewer 4 Report
Comments and Suggestions for Authors
It is an exciting, well-designed study. The authors investigated the affected and infective lesions qualitatively and quantitatively. I have some comments for the authors.
The statistical part will be presented separately. Now, it is after the presentation of Reflected Light and Fluorescence Intensity Calculations.
The authors should describe in more detail how to measure the intensity of reflection.
The authors should discuss the limitations of the study.
Finally, I don't understand the clinical extrapolation of this methodology. In my opinion, the microCT can't used in clinical conditions. OCT is very expensive and has no practical use for the differentiation between affected and infected dentin. Indeed, the authors should discuss this point. On the other hand, in the introduction section, they should focus on why it is important to detect the affected dentin and distinguish it from the infected (minimal invasive dentistry etc).
Author Response
It is an exciting, well-designed study. The authors investigated the affected and infective lesions qualitatively and quantitatively. I have some comments for the authors.
The statistical part will be presented separately. Now, it is after the presentation of Reflected Light and Fluorescence Intensity Calculations. The authors should describe in more detail how to measure the intensity of reflection.
- Detail regarding processing of the intensity calculations for reflectance and fluorescence was moved to the end of sections 2.3 and 2.4 and more detail was provided. The statistics were moved to a separate section, section 2.5.
The authors should discuss the limitations of the study.
- Added line 406:
One limitation of this study is that extracted teeth with visible shadows were not used. This was due to the challenge in collecting a sufficient number of samples. We hope to be able to collect enough extracted teeth with visible shadows for use in a future SWIR imaging study. It would also be useful to examine the areas of infected dentin that had reduced reflectivity at SWIR wavelengths with infrared spectroscopy to determine if there is increased absorption and identify what chemical species are responsible.
Finally, I don't understand the clinical extrapolation of this methodology. In my opinion, the microCT can't used in clinical conditions. OCT is very expensive and has no practical use for the differentiation between affected and infected dentin. Indeed, the authors should discuss this point. On the other hand, in the introduction section, they should focus on why it is important to detect the affected dentin and distinguish it from the infected (minimal invasive dentistry etc).
- Motivation for the study was provided by the very different appearance of an occlusal lesion with a well-defined shadow in SWIR images and OCT versus the visible image in a previous clinical study and the lack of knowledge regarding the appearance of infected and affected dentin at SWIR wavelengths. In that study the large shadow could be seen at visible wavelengths yet the shadow was not visible at SWIR wavelengths and with OCT. These added sentences address this concern and and the potential usefulness of SWIR imaging and OCT. Added at line 376:
We had hoped to observe a large increase in the mean reflectivity of infected dentin compared to sound dentin so that it would be straightforward to use the same approach to remove dentinal lesions. It appears that this is more complex and additional studies are needed to determine if serial SWIR or OCT images taken in real-time during cavity preparations can be used for the selective removal of the deep dentinal lesions investigated in this study.
It is important to explain the clinical significance that the mean reflectivity of affected and sound dentin are similar at SWIR wavelengths while the reflectivity of affected dentin is significantly lower than sound dentin at visible wavelengths. Visible wavelengths cannot be used to determine whether the shadows are caused by affected or infected dentin, however SWIR imaging and OCT can be potentially used to eliminate those shadows caused solely by affected dentin or only by stains. Large areas of affected dentin are unlikely to create subsurface shadows at SWIR wavelengths similar to what is observed at visible wavelengths. There are profound differences in the way reduced light scattering due to the formation of translucent dentin and reduced reflectivity due to absorption by stains translate to the visibility of lesion shadows viewed from the occlusal surface.
Round 2
Reviewer 3 Report
Comments and Suggestions for Authors
Dear Authors, in my opinion after the revision, now the paper can be considered for publication.
Reviewer 4 Report
Comments and Suggestions for Authors
The authors addressed the points of my previous revision. I have no further comments.